# Turning sulfonyl and sulfonimidoyl fluoride electrophiles into sulfur(VI) radicals for alkene ligation

Xing Wu[1,2], Wenbo Zhang[1,2], Guangwu Sun[1], Xi Zou[1], Xiaoru Sang[1], Yongmin He [1] & Bing Gao [1] ✉

Sulfonyl and sulfonimidoyl fluorides are versatile substrates in organic synthesis and medicinal chemistry. However, they have been exclusively used as S(VI)$^+$ electrophiles for defluorinative ligations. Converting sulfonyl and sulfonimidoyl fluorides to S(VI) radicals is challenging and underexplored due to the strong bond dissociation energy of $S^{VI}$–F and high reduction potentials, but once achieved would enable dramatically expanded synthetic utility and downstream applications. In this report, we disclose a general platform to address this issue through cooperative organosuperbase activation and photoredox catalysis. Vinyl sulfones and sulfoximines are obtained with excellent $E$ selectivity under mild conditions by coupling reactions with alkenes. The synthetic utility of this method in the preparation of functional polymers and dyes is also demonstrated.

Compared to the chloride analogues, sulfonyl and sulfonimidoyl fluorides (SFs) are relatively inert due to the high reduction potential and bond strength of $S^{VI}$–F (Fig. 1a)[1,2]. They have exhibited better reaction selectivity and improved stability to heat, hydrolysis, and reduction[3–6]. In addition, SFs are optically stable at the stereogenic sulfur(VI) centers, whereas the chloride analogues are liable to racemization[7–11]. These unique advantages have made SFs useful in synthesis[12–18] and laid the foundation for the recent sulfur(VI) fluoride-exchange (SuFEx) click chemistry[19–22].

Activation of the SFs for fluoride-exchange ligations often requires stringent conditions[23,24]. Vorbrüggen and Gembus found that organosuperbases were able to promote nucleophilic substitution of sulfonyl fluorides by heteroatoms[25,26]. This strategy has been widely used and further developed by several groups[27–32]. Hydrogen bonding was another potential driving force for $S^{VI}$–F activation[33], especially during covalent capture of biomolecules under physiological conditions[19,34–38]. Strong Lewis acids could also abstract fluoride to induce Friedel-Craft-type sulfonylations[39–41] and sulfoximidations[42,43]. However, almost all reported reactions were based on the heterolytic $S^{VI}$–F cleavage mechanism, where SFs served as S(VI)$^+$ electrophiles.

Generating S(VI) radicals from SFs is still underexplored to date, but once achieved would greatly expand their synthetic utility. The excellent balance between thermal stability and kinetic reactivity of SFs makes them superior to many reported S(VI) radical precursors, especially when dealing with complex and challenging systems. For example, sulfonyl chlorides are well known for radical sulfonylations (−1.30 V for $PhSO_2Cl$, Fig. 1a)[44–47], but are often accompanied by hydrolysis and unwanted chlorination products in many reactions (Fig. 1b)[48,49]. The use of sulfinates as radical S(VI) reagents is also limited because of their high sensitivity to oxidants. During the preparation of this manuscript, the Luo and Molander groups reported radical sulfonylations of aryl sulfonyl fluorides[50,51]. However, sulfonimidoyl fluorides were not discussed and a general method for radical ligation of a variety of SFs is still lacking.

Direct homolytic or reductive cleavage of $S^{VI}$–F bonds to S(VI) radicals is challenging due to their high bond strength and reduction potentials (−1.74 V vs SCE for $PhSO_2F$)[2]. In recent studies[52–56], $FSO_2$• radical was generated from $FSO_2Cl$ or related precursors by photocatalysis, but $S^{VI}$–F cleavage to $RSO_2$• was not observed from either the reagents or the products. Inspired by the emerging use of photoredox-based dual catalysis as an appealing approach for inert bond

[1]State Key Laboratory of Chemo/Bio-Sensing and Chemometrics, College of Chemistry and Chemical Engineering, Hunan University, Changsha 410082, China. [2]These authors contributed equally: Xing Wu and Wenbo Zhang. ✉e-mail: gaobing@hnu.edu.cn

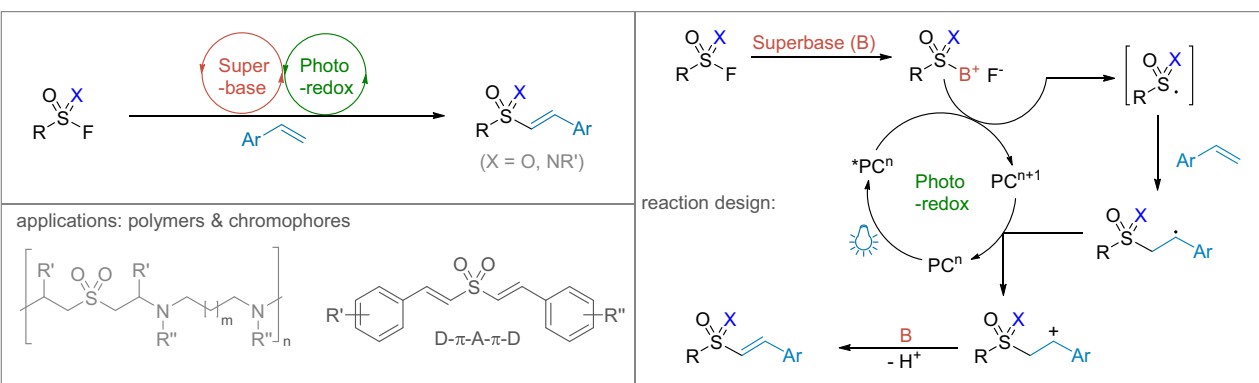

**a. chemical properties of sulfonyl and sulfonimidoyl fluorides vs their chloride analogues (X = O, NR')**

- $D_0^o$ (S–Cl) = 43 kcal/mol
- $D_0^o$ (S–F) = 84 kcal/mol

$E_{red}$ = -1.30 V (vs SCE, R=Ph, X=O)

- liable to hydrolysis and reduction
- liable to racemization at S(VI) when X = NR'
- poor selectivity in reactions (Cl$^{+/\cdot}$ side reaction)

$E_{red}$ = -1.74 V (vs SCE, R=Ph, X=O)

- stabile toward moisture, heat, and reduction
- optically stable at S(VI) when X = NR'
- high selectivity and specificity in reactions

**b. ligations of sulfonyl and sulfonimidoyl fluorides:  nucleophilic mechanism (well-established)  vs  radical mechanism (no general method)**

- challenge: high reduction potential ($E_{red}$) / high bond strength $D_0$ (S$^{VI}$–F)

**c. a general platform for converting sulfonyl and sulfonimidoyl fluorides into S(VI) radicals (this work)**

applications: polymers & chromophores

D-π-A-π-D

reaction design:

**Fig. 1 | The conventional transformations of sulfonyl and sulfonimidoyl fluorides and our reaction design. a** The properties of sulfonyl and sulfonimidoyl halides (F, Cl). **b** Different mechanisms of activating SFs. **c** Our S(VI) radical ligation platform.

functionalization[57–62], we envisioned that a cooperative strategy might be worth trying. We hypothesized that intermediates RSO(X)[B]$^+$ derived from the organosuperbase activation of SFs could be converted in situ to related S(VI) radicals upon visible light irradiation in the presence of photocatalyst, which were then trapped by substrates to complete a reaction cycle (Fig. 1c)[63,64]. Herein, we report that this protocol is generally applicable for converting SFs electrophiles to S(VI) radicals in the reaction with alkenes.

## Results

We initiated the study using phenyl sulfonyl fluoride **1a** and styrene **2a** as model substrates and Ru(bpy)$_3$Cl$_2$ as photocatalyst. And representative results were summarized in Fig. 2 (see Supplementary Information for details). Under the blue LED illumination with 1,8-diazabicyclo[5.4.0]undec-7-ene (DBU) as an additive in dry CH$_3$CN, nearly quantitative sulfonylation product **3aa** was obtained in absolute *E* configuration at room temperature (entry 1). Both visible light and the photocatalyst were required for this reaction (entry 2–3). Without or with less than 3.5 eq. of DBU, no or incomplete conversion of **1a** was observed (entry 4–5). There was no reaction using inorganic bases such as Cs$_2$CO$_3$ (entry 6). Et$_3$N was an effective sacrificial reduction reagent in promoting many photoredox reactions[65–67]. However, it did not work for this reaction (entry 7). Only organosuperbases with similar p*K*b values to DBU were effective[68], including DBN, BTMG and

MTBD (entry 8−10). Ru(bpy)$_3$Cl$_2$ was found optimal over other photocatalysts at a 1.5 mol% loading (entry 11−13). Considering the solvents, the use of DMSO also gave a quantitative yield of **3aa**, while the less polar DCE resulted in no conversion (entry 14−15). CH$_3$CN was used in the following studies because of its low boiling point and ease for handling. Sulfonyl chloride was investigated as a substitute for **1a**, but decomposed completely to give only a trace amount of **3aa** (entry 16). During the reaction optimization, the light on/off experiment was also performed with a time interval of 2 hours. We found that the reaction stopped completely when the light was turned off, but recovered when the blue LED was turned on (Fig. 2, bottom right). This result was helpful in elucidating the reaction mechanism and was further discussed in the mechanism study part of this manuscript.

Using the established conditions, the scope of aryl sulfonyl fluorides was investigated (Fig. 3). Substrates bearing either electron-donating or withdrawing substituents on arenes *para* to the sulfonyl fluoride gave sulfonylation products in high yields (**3aa**–**3ja**). However, the reaction rate was significantly accelerated by electron-deficient substituents, allowing the reaction to be completed within minutes (**3da**, **3fa**, **3ia**). Substrates bearing *meta* or *ortho* substituents also gave satisfactory yields (**3ka**–**3oa**). The mild conditions tolerated functional groups such as halides (**3ha**, **3ma**), cyanide (**3ia**), and carboxylic esters (**3ja**) that allowed for post-functionalization.

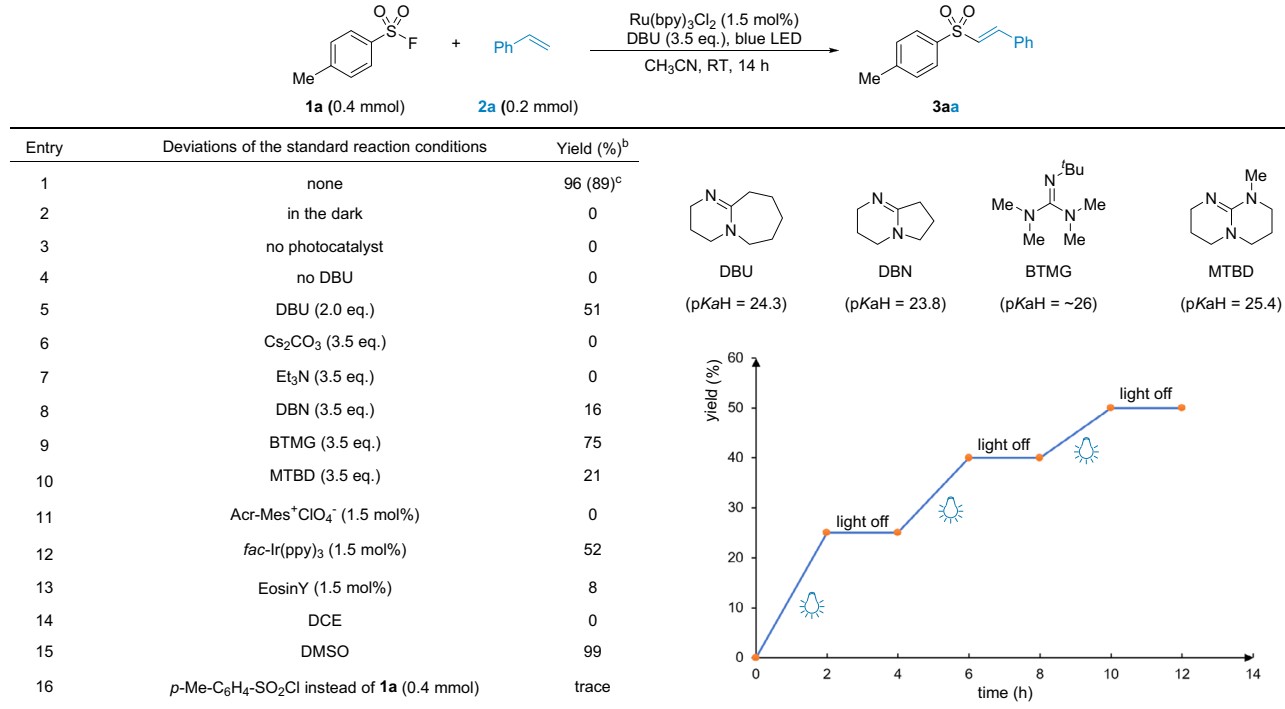

**Fig. 2 | Reaction development.** [a]Standard reaction conditions: **1a** (0.4 mmol), **2a** (0.2 mmol), Ru(bpy)$_3$Cl$_2$ (1.5 mol%), DBU (0.7 mmol), dry CH$_3$CN (2.0 mL), blue LED (5 W), RT, N$_2$ atmosphere. [b]Determined by $^1$H NMR. [c]isolated yield.

Heterocycles were ubiquitous and important building blocks in natural and synthetic molecules. The heteroaryl sulfonyl chlorides were often highly reactive and susceptible to decomposition, whereas their fluoride analogues were much more stable for storage and use. In Fig. 3, we highlighted that this method was applicable to a variety of heteroaryl compounds, including coumaran, benzofuran, thiophene, benzothiozole, quinoline, pyridine, 4-quinazolinone, and indazole (**3na**–**3wa**). Further functionalization of drug-related molecules has also been demonstrated, such as the Celecoxib and Neratinib derivatives **3xa** and **3ya**. In all these reactions, vinyl sulfones were obtained in absolute *E* configuration, as confirmed by X-ray analysis of the product **3aa** (CCDC: 2180194).

The scope of alkenes was next investigated with two model sulfonyl fluorides of opposed electron properties (**1a** and **1f**, Fig. 3). One limitation of this method at this stage is that only styrene derivatives work. Aliphatic alkenes were unreactive, probably due to the mismatched redox potentials of related intermediates in the catalytic cycle. Again, the reactions were very fast with the electron-deficient substrate **1f**, while it usually took hours to reach full conversions with the electron-rich substrate **1a**. A standard reaction time of 14 hours was used for the remaining studies for consistency. Yields were good regardless of the electron properties and steric effects of the styrene derivatives (**3fb**–**3fm**). The 1,1-disubstituted alkene (**3fn**) and the 1,2-disubstituted alkene (**3ap**) worked equally well. Notably, a terminal alkene **3fo** was the major product instead of the intramolecular isomer **3fo'**, probably because that it was kinetically favored in the DBU-deprotonation step (see below). The modification of functional molecules and their derivatives was also demonstrated, including Menthol, Thymol, and Galactose (**3fr**–**3ft**).

Divinyl sulfones had unique utility as dyes[69], building blocks[70,71], anti-inflammatory agents, and tumor cell growth inhibitors[72,73]. But there was no reliable synthetic access to them, especially for the unsymmetrical ones. Reported routes installed the vinyl double bond by using aldehyde-based Wittig-Horner reactions, which often gave *E/Z* mixtures[69], or by using multi-step sequences under harsh reaction conditions[71,74]. In a related study, vinyl sulfonyl chlorides were treated with styrene derivatives to afford atom transfer radical addition adducts. An additional step was required to afford divinyl sulfones at high temperatures, accompanied by uncontrollable desulfonylation side reactions[75]. The vinyl sulfonyl fluorides were good Michael acceptors and SuFEx electrophiles. Methods for their preparation and applications were readily available[52,76–78]. Encouraged by the radical aryl sulfonylation, we then became interested in extending the chemistry to vinyl sulfonyl fluorides. In this context, unsymmetrical divinyl sulfones could be prepared.

To our delight, the standard condition was applicable without further optimization (Fig. 4). Divinyl sulfones were obtained in exclusive *E* configuration. A variety of functional groups were tolerated, such as halides (**5bj**–**5dj**, **5lj**, **5mj**), thiol ether (**5fj**), carboxylic esters (**5jj**, **5oj**), and cyanide (**5rd**). Multi-substituted vinyl sulfonyl fluorides also gave the desired products (**5pa**, **5qa**). Styrene derivatives worked equally well, including heteroaryl alkenes (**5sm**, **5sp**) and disubstituted alkene (**5su**). Complex skeletons could be constructed by taking advantage of the distinctive reactivity of multiple SuFEx handles. For instance, a fluorosulfate −OSO$_2$F handle allowed the connection of Norestrone to 4-iodophenol via nucleophilic substitution. The resulting product was converted to vinyl sulfonyl fluoride and further enabled the preparation of **5ta** with our method. The reaction could be extended to aliphatic sulfonyl fluorides, including the primary and secondary alkyl-substituted substrates (**6aa**, **6ba**), but only ~10% yields of products were obtained. This was probably due to the deprotonation of α-methylene by DBU, which induced the decomposition of the sulfonyl fluorides.

During the study of vinyl sulfonyl fluorides, we observed an unexpected reaction pathway. Namely, vinyl sulfonyl fluorides underwent self-condensation in the absence of styrene to give symmetrical divinyl sulfones in low yields (**5aa**, **5bb**, **5hh**, Fig. 4, bottom section). Initially, we thought a styrene intermediate might be generated in situ from the vinyl sulfonyl radical by releasing one molecule of SO$_2$. But styrene was not found in the large-scale synthesis. And the deuterated product **5aa** was also not observed when the reaction of **4a** was carried out in CD$_3$CN. Therefore, styrene was ruled out as a

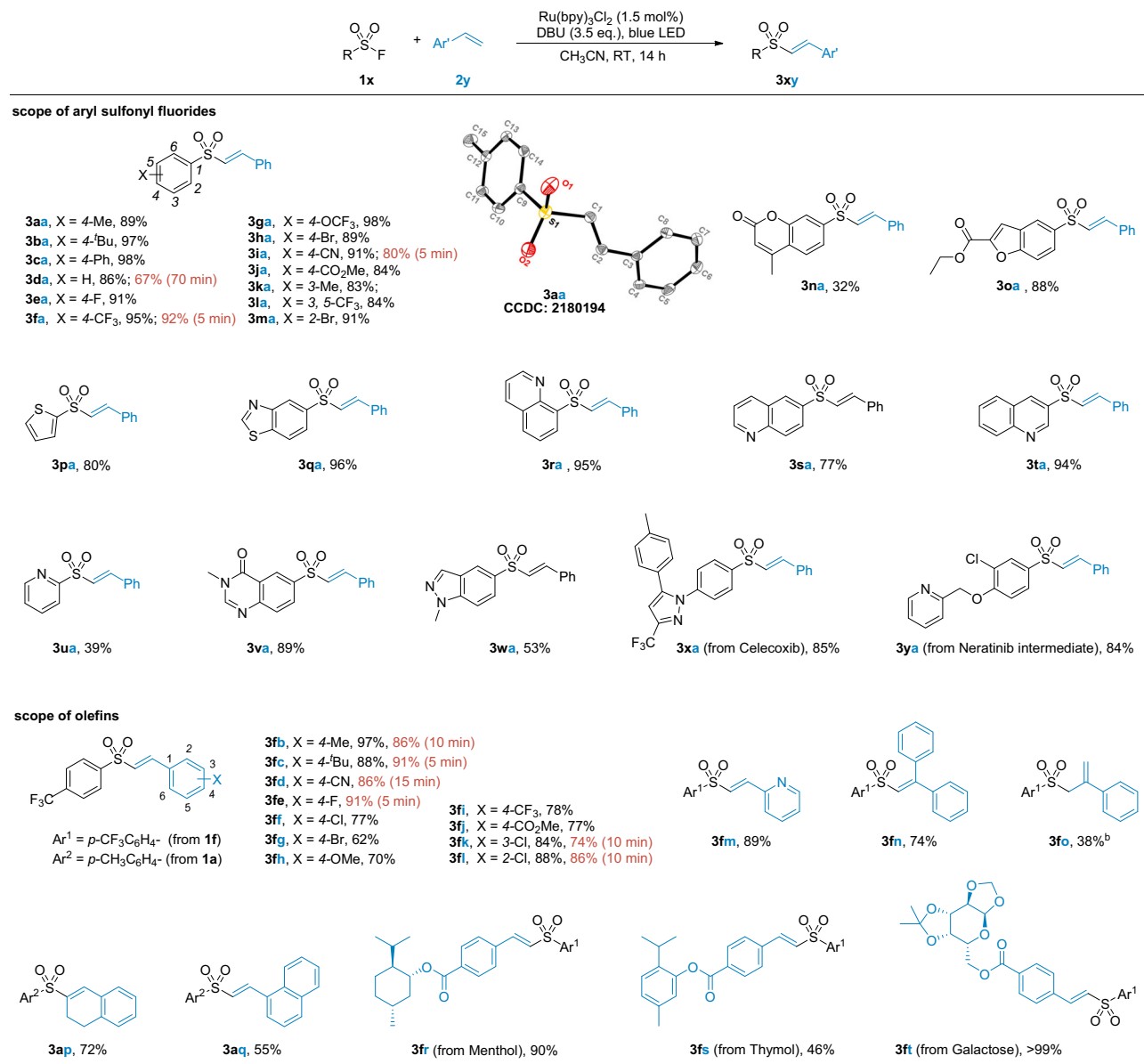

**Fig. 3 | Scope of the aryl sulfonyl fluorides and alkenes.** [a]**1x** (0.4 mmol), **2y** (0.2 mmol), Ru(bpy)$_3$Cl$_2$ (1.5 mol%), DBU (0.7 mmol), CH$_3$CN (2.0 mL), blue LED (5 W), RT, under N$_2$ atmosphere. [b]Linear internal alkene **3fo'** in 8% yield.

potential intermediate. It was likely that the vinyl sulfonyl radical could be directly added the double bond of another vinyl sulfonyl fluoride, resulting in the subsequent ejection of an FSO$_2$• fragment[79] and the final product.

Sulfoximines are useful building blocks in many fields[80–84]. But sulfoximine synthesis via the sulfonimidoyl radical was still rare. To date, only two protocols have been reported, both using sulfonimidoyl chlorides[85,86]. We found that our cooperative activation model was also able to convert sulfonimidoyl fluorides to the corresponding radicals after minor modification of the reaction conditions (see Supplementary Information). As depicted in Fig. 5, imine groups had an important influence on the reaction. Substrates with N-pivaloyl (Piv) group gave the best yield (**8aa**). Their analogues also afforded good yields (**8ab**–**8ae**). However, other N-functional groups such as the tosyl (Ts) were less effective (**8af**), giving mainly the reduction and hydrolysis products. The N-alkyl and N-aryl sulfonimidoyl fluorides were also not good substrates because of their poor electrophilicity[42,43]. The reaction was not sensitive to different electron or steric properties of styrene derivatives. Variations of the

aryl group on sulfonimidoyl fluorides were also investigated and worked quite well. In general, this method is a good complement to the synthesis of vinyl sulfoximines[87–89].

Control experiments were subsequently carried out to gain deeper insight into the mechanism (Fig. 6a). The reaction was inhibited by radical scavengers, such as TEMPO and butylated hydroxytoluene (BHT). Treatment of **1f** with the radical probe cyclopropylstyrene **2t** under standard reaction conditions gave the ring expansion product **9**. Together with the light on/off experiment in Fig. 2, these results supported a photoredox single electron transfer (SET) process, but excluded a radical chain reaction pathway. Besides, sulfonate ArSO$_3^-$ and sulfinate ArSO$_2^-$ salts were excluded as active intermediates, because they did not give any conversion under the standard reaction conditions. The essential cooperative effect of DBU for S$^{VI}$−F activation was further confirmed through a modified experiment based on a previous report[53]. The vinyl sulfonyl fluoride **4a** was obtained from the fluorosulfonylation of styrene under the photoredox reaction conditions. **4a** did not undergo further S$^{VI}$−F reaction in the absence of DBU. However, the

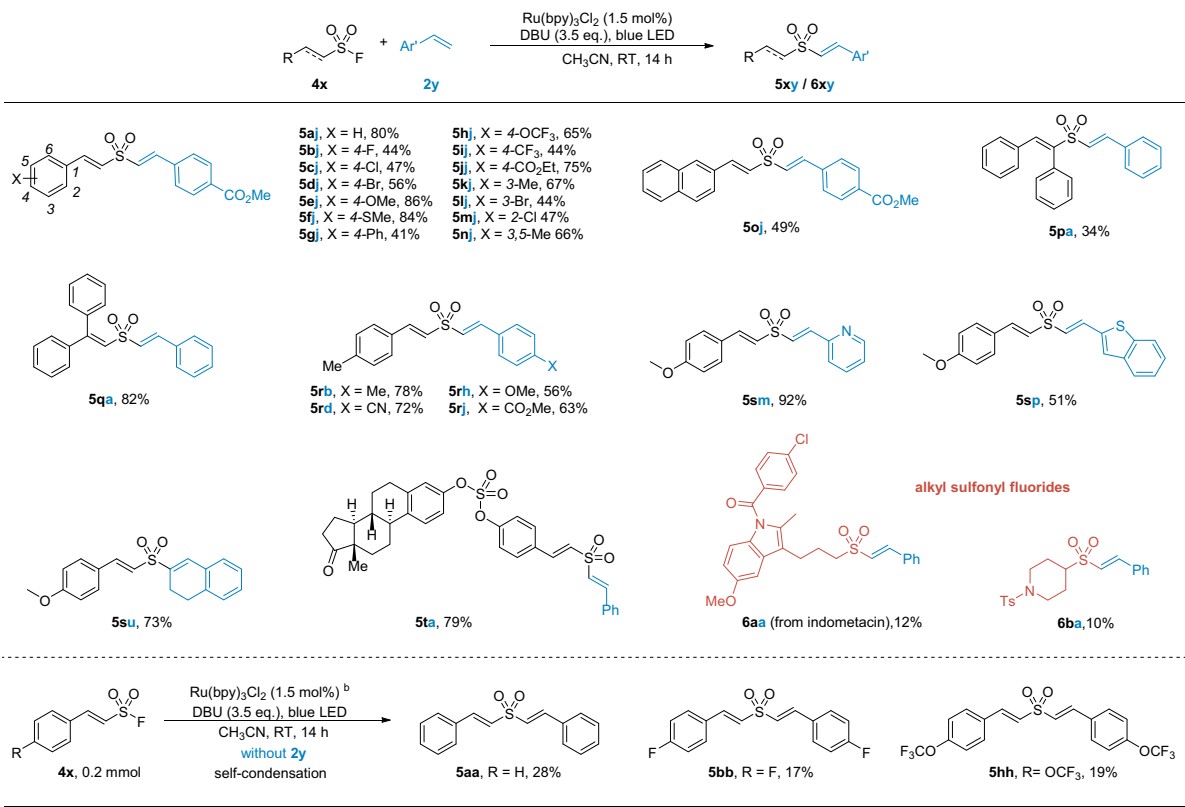

**Fig. 4 | Scope of the vinyl sulfonyl fluorides and alkenes.** [a]**4x** (0.4 mmol), **2y** (0.2 mmol), Ru(bpy)₃Cl₂ (1.5 mol%), DBU (0.7 mmol), CH₃CN (2.0 mL), blue LED (5 W), RT, under N₂ atmosphere. [b]**4x** (0.2 mmol), Ru(bpy)₃Cl₂ (1.5 mol%), DBU (0.7 mmol), CH₃CN (2.0 mL), blue LED (5 W), RT, under N₂ atmosphere.

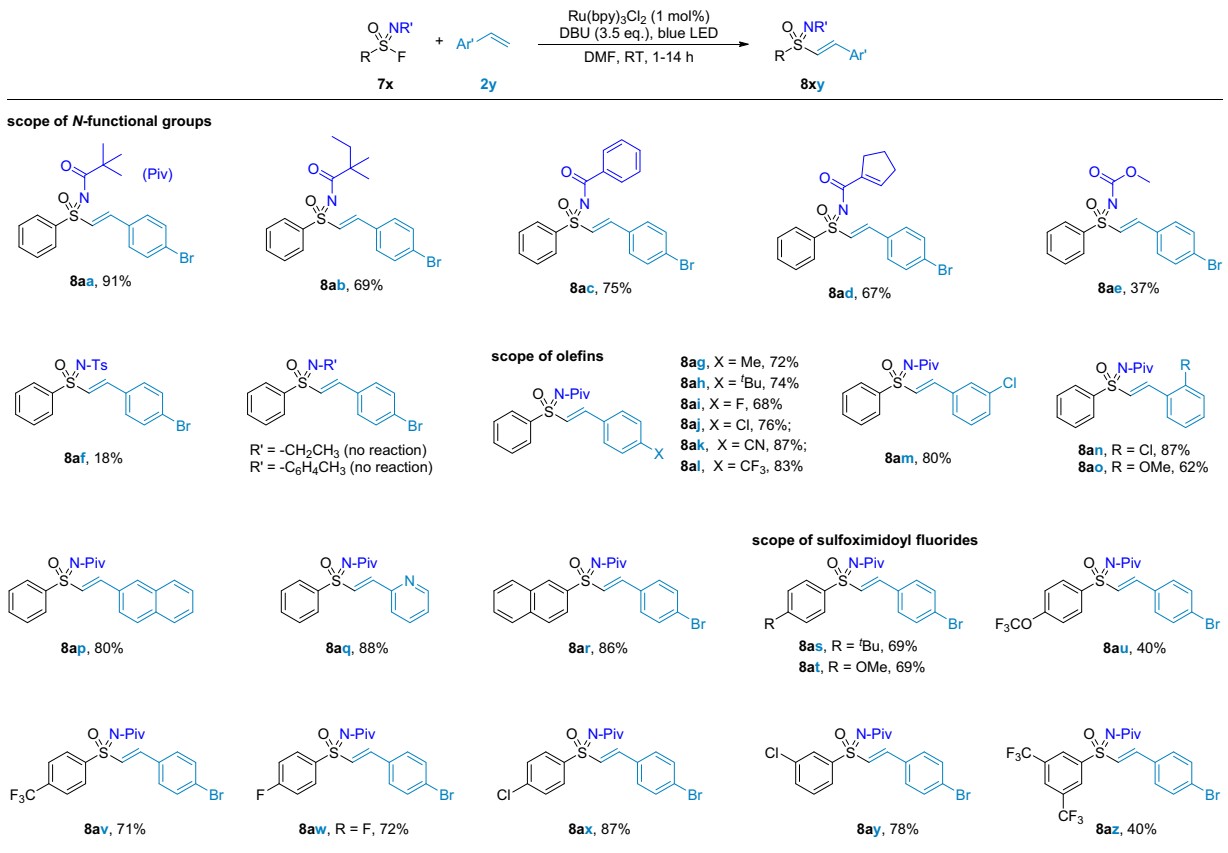

**Fig. 5 | Scope of sulfonimidoyl fluorides.** [a]**4x** (0.4 mmol), **2y** (0.2 mmol), Ru(bpy)₃Cl₂ (1.0 mol%), DBU (0.7 mmol), DMF (2.0 mL), blue LED (24 W), RT, under N₂ atmosphere.

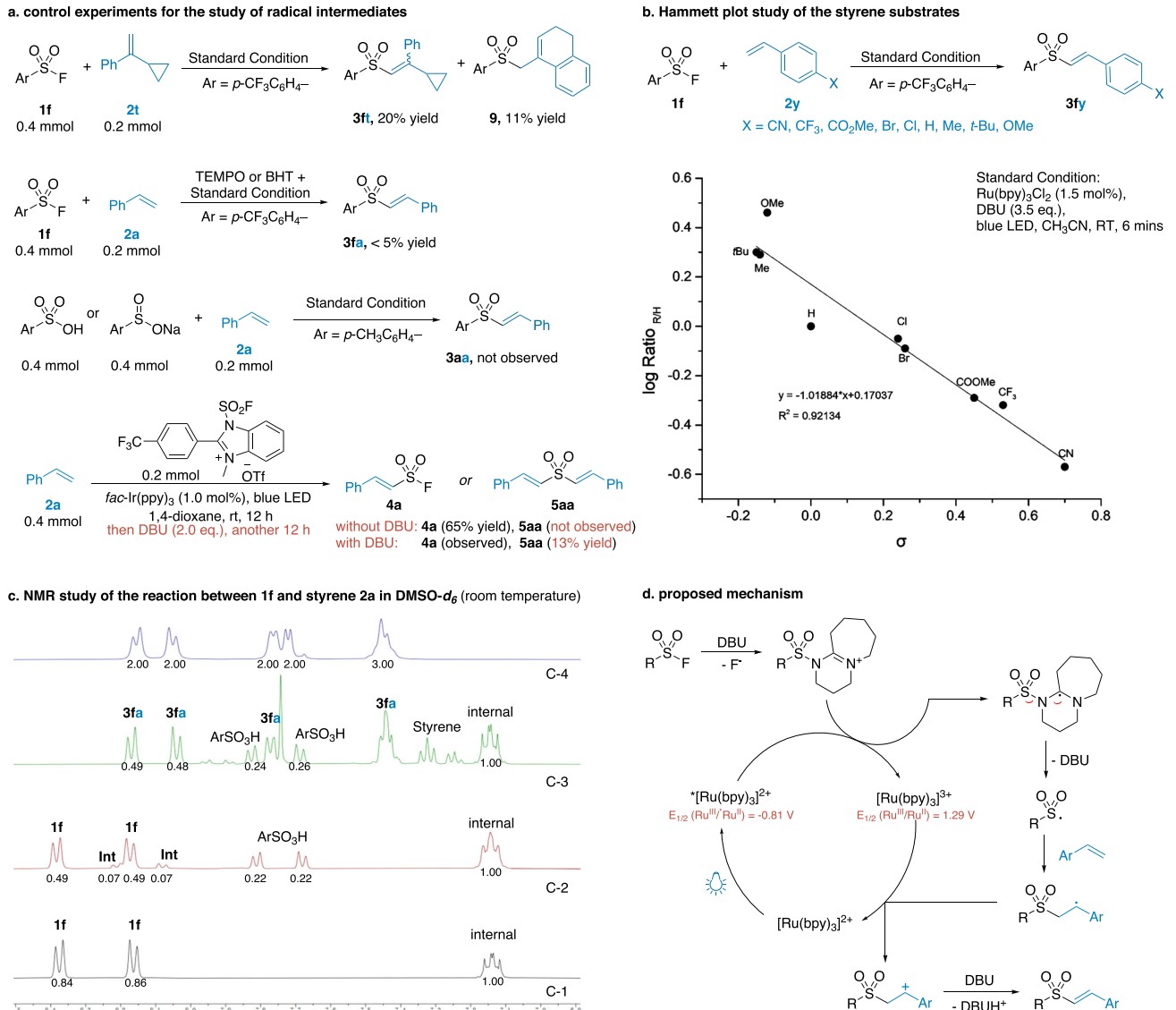

**Fig. 6 | Mechanism study. a** Investigation of radical intermediates. **b** Hammett analysis performed with the styrene substrates. **c** $^1$H NMR study of the reaction progress in DMSO-$d_6$. (Spectrum c-1: mixture of 0.1 mmol of **1f** and 0.1 mmol of 1,3,5-trifluorobenzene as internal standard. Spectrum c-2: 15 min after the addition of 0.2 mmol of DBU. Spectrum c-3: 5 min after the reaction with 0.1 mmol of styrene in the presence of 1.5 mol% Ru(bpy)$_3$Cl$_2$ and blue LED illumination. Spectrum c-4: purified **3fa**). **d** Proposed reaction mechanism.

addition of DBU to the same reaction mixture allowed the spontaneous conversion of **4a** to **5aa** by further reaction with styrene (Fig. 6a, last equation).

A Hammett analysis of the relative reaction rate of the *para*-substituted styrenes indicated that the electron-rich substrates underwent faster reactions than the electron-deficient ones (Fig. 6b). Taken together with the olefin migration observed in product **3fo** (Fig. 3), a benzylic cation intermediate should be involved in the late stage of the reaction cycle. In addition, we learned from Figs. 3 and 4 that the electron-rich sulfonyl fluorides reacted much more slowly than the electron-deficient ones (14 hours for **1a** vs 5 mins for **1f**).

The reaction of sulfonyl fluoride **1f** with styrene was monitored by $^1$H and $^{19}$F NMR in deuterated CH$_3$CN and DMSO (Supplementary Information, Page 66-71). A typical example was depicted in Fig. 6c. Substrate **1f** was partially hydrolyzed to sulfonic acid after 15 min of incubation with DBU in DMSO-$d_6$ at room temperature. This was because a trace amount of water could not be avoided in the deuterated solvents (Fig. 6c, c–2). A set of newly formed peaks **Int** probably

belonged to the proposed intermediate RSO$_2$[DBU]$^+$, which was confirmed by the mass spectrum but could not be isolated due to its high activity (Supplementary Information, Page 69). Upon light irradiation in the presence of styrene and photocatalyst, the remaining sulfonyl fluorides were fully consumed and **3fa** was obtained in 5 min. The **Int** species was also consumed (Fig. 6c, c–3). DBU was quantitatively recovered after the reaction, suggesting that it was not a sacrificial reagent for the photoredox cycle.

In the luminescence quenching experiments, DBU was able to quench the excited Ru(bpy)$_3$Cl$_2$, while other components were not (Supplementary Information, Page 84). Although this suggested that DBU might be directly involved in the early stage of the photoredox cycle, we still thought it unlikely that the reaction would proceed through a reductive quenching cycle for the following reasons. The reduction potential of Ru(II)*/Ru(I) was +0.77 V (vs SCE)[59], which was known for reductive quenching by aliphatic amines such as Et$_3$N (+0.83 V vs SCE) and *i*-Pr$_2$NEt[65–67]. But we did not observe any reaction when Et$_3$N and *i*-Pr$_2$NEt were used during reaction optimization. The

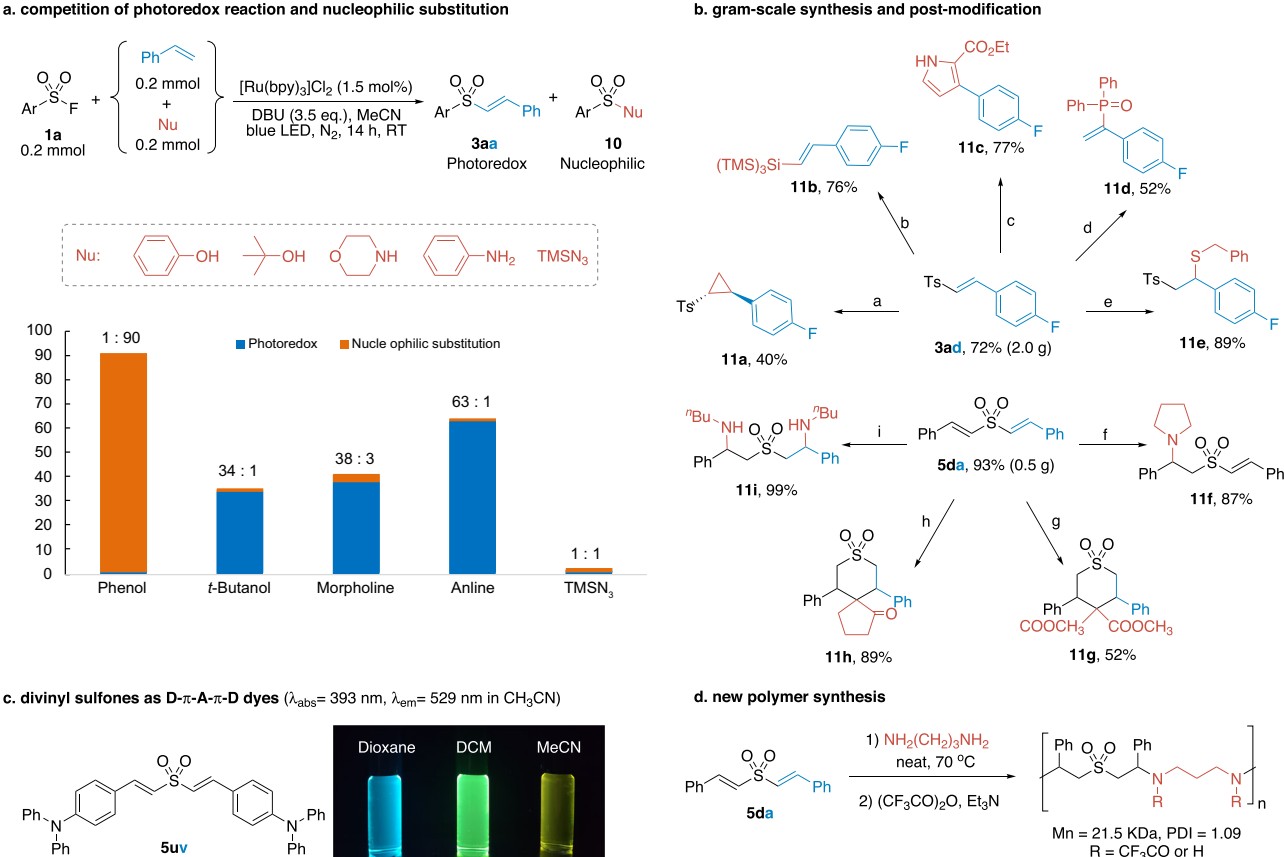

**Fig. 7 | Applications of the radical sulfonylation reaction. a** Competition reaction between the radical and nucleophilic substitution. **b** Gram-scale synthesis and post-modification. (Reaction conditions: [a]trimethylsulfoxonium iodide, NaOH, rt. [b]tris(trimethylsilyl)-silane, AIBN, reflux. [c]ethyl isocyanoacetate, NaOH, rt. [d]DPPO, KOH, $O_2$, rt. [e]benzyl mercaptan, $Et_3N$, rt. [f]pyrrolidine, rt. [g]dimethyl malonate, triton-B, 70 °C. [h]cyclopentanone, triton-B, 70 °C. [i]n-butylamine, 70 °C). **c** Divinyl sulfone as dyes. **d** New polymer synthesis.

potential of DBU was even higher (+1.28 V vs SCE)[90]. Therefore, the reductive quenching of Ru(II)* by DBU did not make sense in this reaction. If this reductive quenching with DBU occurred for whatever reason, it remained challenging for Ru(I) ($E^{II/I} = -1.33$ V vs SCE) to reduce sulfonyl fluorides. Because the reduction potential of $PhSO_2F$ was −1.74 V vs SCE. On the other hand, oxidative quenching of Ru(II)* by sulfonyl fluorides was also unlikely, because the reduction potential of Ru(II)*/Ru(III) was −0.83 V vs SCE. Hydrogen bonding between the sulfonyl fluoride and the in situ generated DBU·H+ was not observed by [1]H and [19]F NMR when the two reagents were incubated in $CD_3CN$ at a 1:2 ratio ($ArSO_2F$ vs DBU, Supplementary Information, Page 73–80). And considering that the experimental reaction rate of the electron-poor aryl sulfonyl fluoride is much faster than that of the electron-rich one, a reaction mechanism based on the hydrogen bonding activation of SFs seemed unlikely. However, we could not rule out other potential hydrogen bonding species as the active intermediate in our reaction.

A plausible reaction mechanism is proposed in Fig. 6d. Nucleophilic activation of sulfonyl fluoride by DBU was very fast at room temperature, especially with the electron-deficient substrates, giving the intermediate $RSO_2[DBU]^+$ by expelling a fluoride anion. This species was highly active and often ended up with hydrolysis to sulfonic acid in the presence of water. But under blue LED irradiation, it was reduced by the exited photocatalyst Ru(II)*. And a sulfonyl radical RSO₂• was subsequently released via intramolecular fragmentation, then added to styrene that gave a benzylic radical. The radical was further oxidized to carbocation by Ru(III), which yielded vinyl sulfone after deprotonation by DBU. As mentioned above, only those

organosuperbases that are effective in catalyzing the nucleophilic substitutions of sulfonyl fluorides could promote this photoredox reaction, such as DBU, BTMG, and MTBD.

We were curious about which ligation pathway the sulfonyl fluoride would be primarily involved when both the heteroatom nucleophile and the alkene were incubated in one reaction flask under the current reaction conditions, the radical sulfonylation or the conventional DBU-promoted nucleophilic substitution. We found the results varied depending on which competing nucleophile was used (Fig. 7a). Nucleophilic substitution dominated in the presence of phenol, giving sulfate in 90% yield. In contrast, radical sulfonylation became the main reaction when *tert*-butanol, morpholine, and aniline were used as nucleophiles. Both the nucleophilic and radical products were present in trace amounts using TMSN₃ as an additive. These results were helpful for evaluating the relative rates of the two ligation pathways and their functional group tolerance.

In Fig. 7b, we demonstrated that our method could be scaled up to prepare grams of vinyl sulfone **3ad** and divinyl sulfone **5da**. The vinyl sulfones were useful in organic synthesis for divergent post-modifications. For instance, cyclopropanation at the double bond of **3ad** yielded **11a**. The sulfonyl motif was also transformable by silylation (**11b**), pyrrolation (**11c**), and phosphinoylation (**11d**). Vinyl sulfones were excellent Michael acceptors for nucleophilic addition by thiols and amines (**11e**, **11f**). It had important applications in the covalent modification of biomolecules in vitro or in vivo. Chemical probes and enzyme inhibitors based on the vinyl sulfone skeleton have been reported[91].

Divinyl sulfones bearing a push-pull structure were useful chromophores with large Stokes shift[69]. Our method provided a

straightforward protocol for their synthesis. Compound **5uv** was a typical example (Fig. 7c). On the other hand, the mono- or di-functionalizations of divinyl sulfones was controllable (**11f**–**11i**). The reaction with aliphatic amine was particularly efficient that gave the disubstituted product in near quantitative yield (**11i**). This reactivity could be utilized for the preparation of new functional polymers[92]. The condensation of **5da** with diamines yielded polyamines with high molecular weight and low dispersity ($M_n$ = 21.5 KDa, PDI = 1.09, Fig. 7d). A systematic study of this polymerization reaction and related applications is underway.

## Discussion

In summary, sulfonyl and sulfonimidoyl fluorides are advantageous over their analogues in synthesis, which have been exclusively used as electrophiles for fluoride-exchange chemical ligations. A general platform for their conversion to radical ligation reagents has been achieved in this manuscript, by combining photoredox catalysis and organosuperbase activation under mild conditions. The reaction with alkenes affords vinyl sulfones and sulfoximines. It also allows for the preparation of new polymeric materials and chromophores. Their synthetic utility has been largely expanded in linkage chemistry and beyond.

## Methods

Under the $N_2$ atmosphere, a 10 mL dry Schlenk tube equipped with a magnetic stirring bar was charged with sulfonyl fluoride, [Ru(bpy)₃]Cl₂, anhydrous solvent, alkene, and DBU. The reaction mixture was stirred at room temperature for 14 hours under blue LED illumination. The resulting mixture was then transferred to a 50 mL round-bottom flask. After removing the solvent, the residue was purified by flash column chromatography to give vinyl sulfones.

## Data availability

The crystallographic data reported in this study have been deposited at the Cambridge Crystallographic Data Centre (CCDC) under deposition numbers 2180194 (**3aa**). They are free of charge via http://www.ccdc.cam.ac.uk/data_request/cif. All other data supporting the findings of this study are available within the article and Supplementary Information files, and are available from the corresponding author upon request.

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

## Acknowledgements

Financial support was provided by National Natural Science Foundation of China (22001065), the Science and Technology Foundation of Hunan Province (2021JJ30090). We thank professor Dr. K. Barry Sharpless at Scripps Research for his helpful discussion, and thank professor Lin Yuan at Hunan University for instrumental support.

## Author contributions

B.G. conceived and directed the project. X.W. and W.Z. performed the synthetic experiments and analyzed the data. X.S. and Y.H. provided support for the cyclic voltammetry experiment. G.S. and X.Z. provided a helpful discussion on reaction development. B.G. wrote the manuscript. X.W. and W.Z. prepared the Supplementary Information and revised the manuscript.

## Competing interests

The authors declare no competing interests.
