## [Peer Review File · Nature Communications]

REVIEWER COMMENTS

Reviewer #1 (Remarks to the Author):

Gao et al. reported a DBU-mediated radical S-C linkage of sulfonyl fluorides and sulfonimidoyl fluorides. Therein, styrene-type carbon partners were used for the efficient trapping of S(VI) radicals to afford a library of vinyl-linked sulfones and sulfoximines via photocatalysis. This manuscript provides a significant S(VI)-C linkage chemistry and can be published in Nature Communication after these issues addressed.

1. Words described in the manuscript are inconsistent with referenes cited, e.g.

1) In Page 1 Lines 20-23, the references 6-10 do not indicate that S(VI) fluorides compared to its chlorides are optically stable. And it is more suitable that the description should be revised as an excellent chemoselectivity of S(VI) fluorides for S-C linkage chemistry.

2) Favorable properties enable sulfonyl fluorides and sulfonimidoyl fluorides unique unilities, but the literatures about sulfonimidoyl fluorides are missing in Refs 11-14.

3) Few highly reactive substrates are used for fluoride-exchange ligations, yet Refs 19-22 describe that S(VI) fluorides are harnessed as an activating reagent to access sulfur-free molecules.

4) Hydrogen bond can be applicable for the activation of S(VI)-F bond in Page 1 Line 33, and the description should be proved via several representative SuFEx examples.

2. The two important papers were missing about the S-C linkage of sulfonyl fluorides via photocatalysis, involving Luo's work (Org. Chem. Front. 2023, 10, 404.) and Molander's work (Org. Lett. 2023, 25, 2084.), where the relevant descriptions and figures should be supplemented.

3. In Page 5 Line 44, imine groups show an important impact on the reaction. It should be illustrated how groups with different electronic properties affect the S(VI)-C linkage, and the relevant references (Nat. Synth. 2022, 1, 455. Angew. Chem. Int. Ed. 2022, 61, e202207100.) need to be noted.

4. Why did compound 3fo afford a disubstituted alkene rather than an unsaturated one? Why did N-Ts sulfonimidoyl fluoride display low linkage efficiency (8f)?

5. The signals about the changes of S-F bond in compound 1f should be shown in Figure S9.

6. The reaction of sulfonyl fluoride with DBU affords the key intermediate (RSO₂[DBU]⁺), which was confirmed via MS. Is there a similar intermediate for sulfonimidoyl fluoride? Zuilhof deemed the process was hard to proceed, due to the steric bulk of DBU (Angew. Chem. Int. Ed. 2020, 59, 7494).

7. The content of Figures 2C (about C-1, C-2, and C-3) and 3C (about the transformation of products) was difficult for the reviewer to understand.

8. Examples of problematic sentences need to be corrected and examined in the manuscript.

1) ..., accordingly, exhibit better selectivity... (Page 1, Lines 16-20)

2) ..., however, did not work... (Page 2, Lines 40-42)

3) This method provided complementary access to diverse vinyl sulfoximines through radical ligation... (Page 5, Line 45)

4) Vinyl sulfonyl fluoride 4a could be produced by radical fluorosulfonylation.. (Page 6, Line 41)

Reviewer #2 (Remarks to the Author):

The manuscript – NCOMMS-23-11185– by Wu et al. describes an application of organosulfur(VI) fluorides – sulfonyl fluorides and sulfonimidoyl fluorides – as radical reagents using photoredox catalysis and a superbases. The transformations focus on sulfonylation of alkenes to make vinyl sulfones and sulfoximines. While sulfonyl fluorides have been demonstrated to serve as radical precursors with boronic acid (ref 51), this work highlights how organic S(VI) fluorides can be used more broadly as radical precursors, expanding from sulfonyl fluorides to sulfoximines. The authors demonstrate the method works with a broad array of electronically and sterically diverse aryl sulfonyl fluorides, sulfonimidoyl fluorides and styrene derivatives. Notably, the method does have limitations in that products using alkyl sulfonyl fluorides are low yielding, and only aryl vinyl coupling partners can be used. However, this limitation does not distract from the impact of this work, because the manuscript provides solid evidence that a broader array of S(VI) fluorides could be used as radical precursors – this work will enable a new branch in S(VI) fluoride chemistry focused on a radical-based approach. The mechanistic study does provide convincing evidence of a radical pathway. The compounds are appropriately characterized, and manuscript was well-written. This paper will have impact across many disciplines in synthetic, biomolecular, and material fields and could be a suitable interest to Nature Communication.

However, there are significant mechanistic questions remaining should be addressed; especially regarding evidence of the proposed DBU adduct. Several mechanistic arguments are hinging on the presence of this adduct as it stands, more experiments are required to provide more convincing evidence of its existence. A considerable portion of this manuscript lies in the mechanistic study, and these issues are preventing recommendation for publication. My recommendation is for revisions, and I would welcome another round of review.

Questions/concerns to be addressed:

1) In page 1, column 2 and in figure 1, the authors state the reduction potential of PhSO₂F is –1.74 vs SCE and used ref 51 as the reference for that value. Although looking at ref 51 and its SI there are no

data suggesting that number. I certainly may have missed this, but perhaps this is the incorrect reference?

2) Title of table 4 should be "Scope of sulfonimidoyl fluorides".

3) Figure 1b suggest that there is "no general method" for sulfonyl fluorides to serve as sulfur-based radicals. Ref 51 is an example to the contrary. The authors amend the figure and statements to reflect this fact. Additionally, it would be appropriate to discuss this paper as well on the first page, second column second paragraph.

4) A piece key mechanistic data is the detection of the $\text{RSO}_2[\text{DBU}]^+$ adduct by mass spectroscopy inferring the role of DBU in activating the S(VI) fluoride toward photocatalysis. These intermediates are indeed unstable as evidence by Gembus et al. , but the analysis of the results is a bit unclear. The authors provided ^1H and ^{19}F NMR data as well as mass spec data to suggest the formation of the $\text{RSO}_2[\text{DBU}]^+$, but some questions remain:

a. The ^{19}F NMR of $\text{pCF}_3\text{PhSO}_2\text{F}$ (1f) plus DBU does not look like there is any appreciable decay of the trifluoromethyl signal of 1f and only a small signal for the proposed DBU adduct. Similarly, with the ^1H NMR spectrum. This is especially challenging since evidence of a ^{19}F NMR standard is not present in Figure S9. Without integrations of the key signals, it is hard to discern the correlation between the new peaks in the ^{19}F and ^1H and that they correlate to the DBU adduct. Integration of these peaks in relation to the standard in both ^{19}F and ^1H spectra would be helpful.

Additionally, a S(VI)-F peak is usually around +50-65 ppm, so evidence that there is quantitative decay of the S-F peak is needed and see if it correlates approximately to the amount of proposed intermediate formed. These data were not provided.

Lastly, if the $\text{RSO}_2[\text{DBU}]^+$ adduct is formed, one would expect evidence of F⁻ anion or hydrogen bonding between the anion and $\text{RSO}_2[\text{DBU}]^+$ adduct (see point b).

b. Instead of the $\text{RSO}_2[\text{DBU}]^+$ adduct, the detected mass could be a strong hydrogen bonding complexation between the sulfonyl fluoride and protonated DBU. On page 7, column two, the authors suggest that since they do not observe hydrogen bonding between the sulfonyl fluoride and protonated DBU, quenching of the excited Ru intermediate could not occur with the hydrogen bonding complex at room temperature. Currently, it is not clear this possibility can be eliminated. Furthermore, it is hard to reconcile with the experiments shown in Figure S8 and S9, that the observed intermediate is operational in the reaction since there is so little of it observed in the experiment.

Due to molecular dynamics at room temperature for NMR spectroscopy, hydrogen bonding can often not be detected at room temperature. Usually in these cases, lower temperatures are needed to observe the HF hydrogen bonding interaction. A suggested experiment would be to allow the sulfonyl fluoride 1F and DBU to stir for longer than the 15 min the experiment – perhaps a few hours – to see if there more of the proposed intermediate detected by NMR spectroscopy. Then taking a ^{19}F and ^1H NMR spectrum at lower temperature (e.g., $-20\text{ }^\circ\text{C}$ or lower) see if one observes the S(VI)–F peak become a doublet. Also, H/F coupling would also show in the ^1H spectrum. This experiment would help to resolve:

- Distinguish between the $\text{RSO}_2[\text{DBU}]^+$ adduct and hydrogen bonding between SF and protonated DBU. If hydrogen bonding between SF and protonated DBU exists, one would observe H/F coupling both in the sulfonyl fluoride signal in the ^{19}F spectrum and in the ^1H NMR spectrum.
- If $\text{RSO}_2[\text{DBU}]^+$ adduct is present, you will see more consumption of the S–F signal in ^{19}F spectrum (using a fluorine based internal standard) and potentially the missing F- peak.
- If more intermediate is formed and the experiments of S8/S9 were redone, it would be more convincing that the observed intermediate (whatever the speciation) is more likely involved in the reaction mechanism.
- In the absence of more evidence, it is challenging to the authors to eliminate any alternative mechanism or speciation.

Reviewer #3 (Remarks to the Author):

Ligation chemistry of SFs (sulfonyl fluorides and sulfonimidoyl fluorides) as a result of fluorine substitution is unique. And sulfur(VI) fluoride exchange (SuFEx) based on SFs has recently emerged as a valuable tool for different synthetic purposes (Ref 15, Sharpless, *Angew. Chem. Int. Ed.*, 2014, 53, 9430). The manuscript by Wu and Gao details extensive work in the field of SuFEx ligations through a different approach. It is true that nucleophilic substitution pathways have been exclusively practiced in previous

use of SFs while radical process is challenging and little explored. Widened utility of SFs through radical ligation with the method described in this paper would be a big step forward.

The paper outlines the reaction design and development. With cooperative activation by organo-superbase and photoredox catalysis, electrophilic SFs are harnessed for radical functionalization of styrene derivatives. The mechanism looks good in supporting the proposed reaction pathway. It is interesting that DBU plays an essential role. This dual activation model should be inspiring for future development of strategies in activating other inert bonds.

The authors demonstrate synthetic applications of their method and products derived therefrom. Given the wide availability of SFs and mild ligation condition, this method should be useful to the synthetic community and beyond. The unsymmetric divinyl sulfones are attractive Michael acceptors either in bioconjugation or synthetic materials.

In summary, the paper provides a nice extension of the SuFEx ligation. The method is useful and the strategy meaningful. SFs are not simple analogues of sulfonyl chlorides or sulfoximidoyl chlorides (SCIs, which are known for radical reactions). They are doing much better than SCIs in terms of stability-reactivity balance. The development of radical ligation based on SFs is more challenging but will be paid off. The work satisfies the novelty requirements necessary for Nature Communication. I strongly recommend publication of this paper.

Additional comments:

- 1) Substrate scope of the sulfur(VI) radical acceptor is relatively narrow and limited to styrene derivatives. Similar S-radicals like Ts radical could react with a wide range of alkenes. Could the authors comment about this? Would it be possible to switch to other alkenes such as acrylate derivatives and even unactivated alkenes?
- 2) Post-transformations in Figure 3C are useful but might not be well-presented in terms of clarity. This part should be improved.
- 3) There is another paper online very recently that deal with radical sulfonylation with aryl sulfonyl fluorides (doi.org/10.1021/acs.orglett.3c00447). I guess these two independent papers were completed and submitted almost at the same time. Synthetic method described in this paper looks strategically different and seems more general, as it also applies to vinyl sulfonyl fluorides and sulfonimidoyl fluorides (these two types of substrates are not discussed in the OL paper, probably not working). Therefore, the novelty and significance of this paper would not be affected. But the OL paper should be cited in the revised manuscript.
- 4) Does this method work for sulfonyl chlorides? It would be interesting to compare the results of these two sulfonyl sources.
- 5) In the proposed mechanism, a carbocation intermediate is formed. Thus there is an opportunity to develop alkene difunctionalization reactions through trapping the cation intermediate with nucleophiles.

Manuscript Information

Manuscript title:

**Turning Sulfonyl and Sulfonimidoyl Fluoride Electrophiles into Sulfur(VI) Radicals for
Alkene Ligation**

Manuscript ID (*previous*): **NCOMMS-23-11185**

Response to Reviewer 1:

Reviewer 1: Gao et al. reported a DBU-mediated radical S-C linkage of sulfonyl fluorides and sulfonimidoyl fluorides. Therein, styrene-type carbon partners were used for the efficient trapping of S(VI) radicals to afford a library of vinyl-linked sulfones and sulfoximines via photocatalysis. This manuscript provides a significant S(VI)-C linkage chemistry and can be published in Nature Communication after these issues addressed.

1. Words described in the manuscript are inconsistent with referenes cited, e.g.

1) In Page 1 Lines 20-23, the references 6-10 do not indicate that S(VI) fluorides compared to its chlorides are optically stable. And it is more suitable that the description should be revised as an excellent chemoselectivity of S(VI) fluorides for S-C linkage chemistry.

Our Response: We thank the reviewer for his/her valuable comments. We have replaced Ref 6 (in the previous manuscript) with Ref 7 (*J. Org. Chem.*, **1979**, *44*, 2061, in the revised manuscript). The poor optical stability of chiral sulfonimidoyl chloride was claimed by Prof. Johnson in Ref 7, quote: “*The optically active N-methylbenzenesulfonimidoyl chloride (3) was obtained by oxidation in ether at -78 °C, but 3 was not isolated because racemization took place very fast at higher temperatures*”. Sulfonyl fluorides are more optically stable according to the experimental data in Refs 8-11 (in the revised manuscript).

Reviewer 1: 2) Favorable properties enable sulfonyl fluorides and sulfonimidoyl fluorides unique unilities, but the literatures about sulfonimidoyl fluorides are missing in Refs 11-14.

Our Response: Thanks. Refs 11-14 (in the previous manuscript, now Refs 12-15 in the revised edition) are nice accounts of sulfonyl fluorides. Sulfonimidoyl fluorides were also mentioned, but without systematic discussion. So, we have added Refs 16-18 to the revised manuscript. These are representative examples of synthetic applications of sulfonimidoyl fluorides.

Reviewer 1: 3) Few highly reactive substrates are used for fluoride-exchange ligations, yet Refs 19-22 describe that S(VI) fluorides are harnessed as an activating reagent to access sulfur-free molecules.

Our Response: Thanks. In the previous manuscript, we wanted to point out that most sulfur(VI) fluorides required a stringent activation method. And there were also a few exceptions that needed

to be mentioned, such as some highly reactive substrates in Refs 18-22 (in the previous manuscript). We now agree with the reviewer that desulfonylation reactions are less relevant for the “fluoride exchange ligations”. Therefore, these references and the relevant discussions have been removed in the revised manuscript.

Reviewer 1: 4) Hydrogen bond can be applicable for the activation of S(VI)-F bond in Page 1 Line 33, and the description should be proved via several representative SuFEx examples.

Our Response: Hydrogen bonding has been considered as an important driving force for SuFEx ligations of biomolecules in living systems. Representative examples and relevant discussions can be found in Refs 19, 33-37 (in the revised manuscript).

Reviewer 1: 2. The two important papers were missing about the S-C linkage of sulfonyl fluorides via photocatalysis, involving Luo’s work (Org. Chem. Front. 2023, 10, 404.) and Molander’s work (Org. Lett. 2023, 25, 2084.), where the relevant descriptions and figures should be supplemented.

Our Response: Luo’s work was cited as Ref 51 in the previous manuscript. Molander’s work was online after the submission of this manuscript to *Nat. Commun.* In the revised manuscript, these two papers have been cited as Refs 49,50 at the end of paragraph 3. The relevant comments have also been added.

Reviewer 1: 3. In Page 5 Line 44, imine groups show an important impact on the reaction. It should be illustrated how groups with different electronic properties affect the S(VI)-C linkage, and the relevant references (Nat. Synth. 2022, 1,455. Angew. Chem. Int. Ed. 2022, 61, e202207100.) need to be noted.

Our Response: As suggested by the reviewer, the effects of different *N*-functional groups on the reactivity of relevant substrate has been discussed in the revised manuscript. (Page 5, right column, line 40-44). The two papers (*Nat. Synth.* 2022, 1, 455; *Angew. Chem. Int. Ed.* 2022, 61, e202207100) have been cited as Refs 41 and 42 (in both the previous and revised manuscript), and have been noted therein.

Reviewer 1: 4. Why did compound 3fo afford a disubstituted alkene rather than an unsaturated one? Why did N-Ts sulfonimidoyl fluoride display low linkage efficiency (8f)?

Our Response:

1) The unsaturated internal alkene **3fo'** should be the thermodynamic product (isomer of 3fo, SI, Page S44-S45). However, it was observed as a minor product in very low yield (8%). We hypothesized that the deprotonation of the related benzylic cation intermediate by DBU was less kinetically favored at the α -position of the sulfonyl group due to steric hinderance. The terminal methyl group is more open to attack though it was less acidic.

2) Most of the *N*-Ts functionalized sulfonimidoyl fluoride decomposed to sulfonamide and sulfonamide by reduction or hydrolysis under the standard reaction conditions.

Reviewer 1: 5. The signals about the changes of S-F bond in compound 1f should be shown in Figure S9.

Our Response: Thanks. These spectra have been updated in the revised SI. And we have also added the integrations of the ¹H and ¹⁹F peaks in spectra S9, S10, S11, S12 (SI, Page S66-71), as suggested by the second reviewer. Please see Page 11-14 of this document for details of data analysis if you are also interested.

Reviewer 1: 6. The reaction of sulfonyl fluoride with DBU affords the key intermediate (RSO₂[DBU]⁺), which was confirmed via MS. Is there a similar intermediate for sulfonimidoyl fluoride? Zuilhof deemed the process was hard to proceed, due to the steric bulk of DBU (Angew. Chem. Int. Ed. 2020, 59, 7494).

Our Response:

- 1) Yes, we have also observed similar intermediates for the sulfonimidoyl fluorides by MS, including an electron-rich aromatic substrate (R=^tBu) and an electron-deficient one (R=Cl). The spectra are listed below. These data have been added to the revised SI (Page S83).
- 2) Zuilhof's reaction is different. Their conclusion, based on their experimental and computational

results, is that “activation of S-F loss by the base has been one of the most widely accepted mechanism models to date. This might be the case for much poorer nucleophiles, including the Si-protected phenols, but such activation is apparently not needed for phenolate SuFEx reactions.” In other words, DBU may not have a chance to substitute for S-F in the presence of phenolate reagent. But the substitution by DBU is still likely to occur in the absence of strong nucleophile.

By the way, Doyle group have also observed a similar DBU-substituted intermediate by MS in their reactions (ArSO₂[DBU]⁺, *J. Am. Chem. Soc.* **2015**, *137*, 9571–9574, Ref 26 in the revised manuscript).

Figure S23. MS (ESI, Shimadzu) monitoring of the reaction mixture of sulfonimidoyl fluoride **7s** and DBU in DMF (ArSO[=NR][DBU]⁺ species from **7s** and DBU was detected).

Figure S24. MS (ESI, Shimadzu) monitoring of the mixture of sulfonimidoyl fluoride **7x** and DBU in DMF ($\text{ArSO}[\text{=NR}][\text{DBU}]^+$ species from **7x** and DBU was detected).

Reviewer 1: 7. The content of Figures 2C (about C-1, C-2, and C-3) and 3C (about the transformation of products) was difficult for the reviewer to understand.

Our Response: Thanks. The integrations of the relevant peaks have been added to the spectra in Figure 2C. And the content has been slightly modified. A full description of the content of Figure 2C can be found in the revised manuscript (page 6, lines 45-46 and page 7, lines 1-6) and SI (pages S66-S71). The content of Figure 3C has been reorganized.

Reviewer 1: 8. Examples of problematic sentences need to be corrected and examined in the manuscript.

- 1) ..., accordingly, exhibit better selectivity... (Page 1, Lines 16-20)
 - 2) ..., however, did not work... (Page 2, Lines 40-42)
 - 3) This method provided complementary access to diverse vinyl sulfoximines thought radical ligation... (Page 5, Line 45)
 - 4) Vinyl sulfonyl fluoride **4a** could be produced by radial fluorosulfonylation.. (Page 6, Line 41)
- Our Response:** Thanks. We have revised the whole manuscript and improved the languages according to your suggestions.

Response to Reviewer 2:

Reviewer 2: The manuscript – NCOMMS-23-11185– by Wu et al. describes an application of organosulfur(VI) fluorides – sulfonyl fluorides and sulfonimidoyl fluorides – as radical reagents using photoredox catalysis and a superbase. The transformations focus on sulfonylation of alkenes to make vinyl sulfones and sulfoximines. While sulfonyl fluorides have been demonstrated to serve as radical precursors with boronic acid (ref 51), this work highlights how organic S(VI) fluorides can be used more broadly as radical precursors, expanding from sulfonyl fluorides to sulfoximines. The authors demonstrate the method works with a broad array of electronically and sterically diverse aryl sulfonyl fluorides, sulfonimidoyl fluorides and styrene derivatives. Notably, the method does have limitations in that products using alkyl sulfonyl fluorides are low yielding, and only aryl vinyl coupling partners can be used. However, this limitation does not distract from the impact of this work, because the manuscript provides solid evidence that a broader array of S(VI) fluorides could be used as radical precursors – this work will enable a new branch is S(VI) fluoride chemistry focused on a radical-based approach. The mechanistic study does provide convincing evidence of a radical pathway. The compounds are appropriately characterized, and manuscript was well-written. This paper will have impact across many disciplines in synthetic, biomolecular, and material fields and could be a suitable interest to Nature Communication.

However, there are significant mechanistic questions remaining should be addressed; especially regarding evidence of the proposed DBU adduct. Several mechanistic arguments are hinging on the presence of this adduct as it is stands, more experiments are required to provide more convincing evidence of its existence. A considerable portion of this manuscript lies in the mechanistic study, and these issues are preventing recommendation for publication. My recommendation is for revisions, and I would welcome another round of review.

Questions/concerns to be addressed:

1) In page 1, column 2 and in figure 1, the authors state the reduction potential of PhSO₂F is – 1.74 vs SCE and used ref 51 as the reference for that value. Although looking at ref 51 and its SI there are no data suggesting that number. I certainly may have missed this, but perhaps this is the

incorrect reference?

Our Response: We thank the reviewer for the valuable comments and apologize for any inconvenience caused by the citation issue. Our measured reduction potential of PhSO₂F is -1.74 V *vs* SCE (SI, page S85). And the data reported by a Canadian group is -1.7 V *vs* SCE (*E_{p/2}*, Ref 2 in the revised manuscript). Molander group have also reported their experimental data, which is -1.87 V *vs* SCE (Ref 50 in the revised manuscript, *Org. Lett.* **2023**, 25, 2084-2087).

Reviewer 2: 2) Title of table 4 should be “Scope of sulfonimidoyl fluorides”.

Our Response: Thank you. We have corrected the title of Table 4 according to your suggestions.

Reviewer 2: 3) Figure 1b suggest that there is “no general method” for sulfonyl fluorides to serve as sulfur-based radicals. Ref 51 is an example to the contrary. The authors amend the figure and statements to reflect this fact. Additionally, it would be appropriate to discuss this paper as well on the first page, second column second paragraph.

Our Response: Thanks. The Luo and Molander groups provide elegant examples of activating aryl sulfonyl fluorides for sulfonylation reactions through radical intermediates (Ref 51 in the previous manuscript, and Refs 49, 50 in the revised manuscript). However, sulfonimidoyl fluorides are not discussed in their reports. In Figure 1B of the previous manuscript, we were discussing about the lack of a method that was generally applicable to both sulfonyl fluorides and sulfonimidoyl fluorides (X = O, NR). That’s why the comment “no general method” was used. In the revised manuscript, we have modified Figure 1 and updated the corresponding comments at the end of paragraph 3.

Reviewer 2: 4) A piece key mechanistic data is the detection of the RSO₂[DBU]⁺ adduct by mass spectroscopy inferring the role of DBU in activating the S(VI) fluoride toward photocatalysis. These intermediates are indeed unstable as evidence by Gembus et al. , but the analysis of the results is a bit unclear. The authors provided ¹H and ¹⁹F NMR data as well as mass spec data to suggest the formation of the RSO₂[DBU]⁺, but some questions remain:

a. The ¹⁹F NMR of *p*-CF₃PhSO₂F (**1f**) plus DBU does not look like there is any appreciable decay of the trifluoromethyl signal of **1f** and only a small signal for the proposed DBU adduct. Similarly, with the ¹H NMR spectrum. This is especially challenging since evidence of a ¹⁹F NMR standard is

not present in Figure S9. Without integrations of the key signals, it is hard to discern the correlation between the new peaks in the ^{19}F and ^1H and that they correlate to the DBU adduct. Integration of these peaks in relation to the standard in both ^{19}F and ^1H spectra would be helpful.

Additionally, a S(VI)–F peak is usually around +50-65 ppm, so evidence that there is quantitative decay of the S–F peak is needed and see if it correlates approximately to the amount of proposed intermediate formed. These data were not provided.

Lastly, if the $\text{RSO}_2[\text{DBU}]^+$ adduct is formed, one would expect evidence of F^- anion or hydrogen bonding between the anion and $\text{RSO}_2[\text{DBU}]^+$ adduct (see point b).

Our Response:

1) The NMR spectra of S–F have been added to the revised SI. And the integrations of the ^1H and ^{19}F peaks in relation to the internal standard 1,3,5-trifluorobenzene have been updated in spectra S9, S10, S11, S12 (SI, Page S67-S71). These spectra are also appended in the next few pages (Page 1114).

However, the measured intensities of the ^{19}F peaks do not match the theoretical/real values of the related compounds in the mixtures. For example, compound **1f** ($p\text{-CF}_3\text{C}_6\text{H}_4\text{SO}_2\text{F}$) has a $-\text{CF}_3$ group and a $-\text{SO}_2\text{F}$ motif. But the integrated ^{19}F ratio of this compound is always far from 3:1 (CF_3 vs SO_2F). And their relative ratios to the internal standard (1,3,5-trifluorobenzene, -109.23 ppm) are also not in agreement with to the theoretical values. This is probably due to the sensitivity issue of our NMR instrument when using the ^{19}F probe. In contrast, the integrations of the ^1H NMR spectra are accurate and could be used for quantitative analysis. The ^{19}F NMR spectra are not suitable for quantitative analysis, but are fine for qualitative evaluation.

Data analysis of the NMR spectra (Figures S8-S12 in the SI, Page S67-S71):

#a, After 15 min of incubation with DBU, the amount of substrate **1f** decreased to about 60% in CD_3CN (Figure S8) and 58% in $d^6\text{-DMSO}$ (Figure S11). And the rest of **1f** was completely consumed after the photoredox reactions.

#b, After 15 min of incubation of DBU with substrate **1f**, the **Int** species was generated (about 11% yield in CD_3CN and 8% yield in $d^6\text{-DMSO}$, based on the ^1H NMR spectra in Figure S8, S11).

The corresponding ^{19}F signals of the $-\text{CF}_3$ group were found on the ^{19}F NMR spectra (63.9 ppm in CD_3CN and -62.1 ppm in d^6 -DMSO, no $-\text{SO}_2\text{F}$ peak, Figure S9, S12). This species disappeared completely after the photoredox reactions.

#c, After 15 min of incubation of DBU with substrate **1f**, the ArSO_3H was also generated (about 26% yield in CD_3CN and 26% yield in d^6 -DMSO, based on the ^1H NMR spectra in Figure S8, S11). The corresponding ^{19}F signals of the $-\text{CF}_3$ group were found on the ^{19}F NMR spectra (-63.0 ppm in CD_3CN and -61.2 ppm in d^6 -DMSO, no $-\text{SO}_2\text{F}$ peak, Figure S9, S12). The yields of the ArSO_3H further increased after the photoredox reactions (42% yield in CD_3CN and 30% yield in d^6 -DMSO, Figure S8, S11).

#d, Product **3fa** was generated after the photoredox reaction. The yields were about 61% in CD_3CN and 58% in d^6 -DMSO (Figure S8, S11).

#e, There were also few other unidentified byproducts formed after the photoredox reaction. Their yields could not be determined due to the low intensities.

Conclusions based on the data analysis:

#a, The hydrolysis of ArSO_2F to ArSO_3H is the main side reaction in the reaction system. It occurs mainly in the early stage by DBU activation, and continues throughout the reaction process.

#b, The **Int** species and the rest of substrate **1f** (after 15 min of incubation of **1f** with DBU) together were the sulfonyl source of the product **3fa**, the newly formed ArSO_3H , and other unidentified by-products (after the photoredox reaction). These data in Figures S8-S12 alone do not support that **Int** must be the active key intermediate from substrate to product, but they also cannot rule it out. Taken together with other experimental results, we still believe that it is likely to be the $[\text{ArSO}_2\text{DBU}]^+$ species and is responsible as the key intermediate.

2) the F^- anion formed after the incubation of DBU with **1f** was not observed by the room temperature NMR. But it was detected by the low-temperature NMR analysis, which is very likely a $[\text{F}-\text{H}-\text{F}]^-$ species (^{19}F NMR: δ -146.16 ppm, d, $J = 124$ Hz; ^1H NMR: δ 16.01 ppm, t, $J = 128$ Hz, in CD_3CN , Figure S13 and S14, Page 73 and 74 of the SI).

(Note: Figure S8, S9, S11, S12 are appended on the next few pages)

Figure S8. ¹H NMR monitoring of the reaction of **1f** and styrene in CD₃CN (top: δ 8.5 – 1.0 ppm; bottom: δ 8.4 – 6.6 ppm)

Figure S9. ¹⁹F NMR monitoring of the reaction of **1f** and styrene in CD₃CN (top: δ 70.0 – -120 ppm; bottom: left δ 66.0 – 63.5 ppm, middle δ -62.5 – -64.5 ppm, right δ -108.5 – -110.0 ppm)

Figure S11. ^1H NMR monitoring of the reaction of **1f** and styrene in $\text{DMSO-}d_6$ (top: δ 8.7 – 1.0 ppm; bottom: δ 8.5 – 6.6 ppm)

Figure S12. ^{19}F NMR monitoring the reaction of **1f** and styrene in $\text{DMSO-}d^6$ (top: 8 70.0 – -120 ppm; bottom: left 8 66.0 – 64.0 ppm, middle 8 -60.9 – -62.7 ppm, right 8 -103.0 – -112.0 ppm)

Reviewer 2: b. Instead of the $\text{RSO}_2[\text{DBU}]^+$ adduct, the detected mass could be a strong hydrogen bonding complexation between the sulfonyl fluoride and protonated DBU. On page 7, column two, the authors suggest that since they do not observe hydrogen bonding between the sulfonyl fluoride and protonated DBU, quenching of the excited Ru intermediate could not occur with the hydrogen bonding complex at room temperature. Currently, it is not clear this possibility can be eliminated. Furthermore, it is hard to reconcile with the experiments shown in Figure S8 and S9, that the observed intermediate is operational in the reaction since there is so little of it observed in the experiment.

Due to molecular dynamics at room temperature for NMR spectroscopy, hydrogen bonding can often not be detected at room temperature. Usually in these cases, lower temperatures are needed to observe the HF hydrogen bonding interaction. A suggested experiment would be to allow the sulfonyl fluoride ^{19}F and DBU to stir for longer than the 15 min the experiment – perhaps a few hours – to see if there more of the proposed intermediate detected by NMR spectroscopy. Then taking a ^{19}F and ^1H NMR spectrum at lower temperature (e.g., $-20\text{ }^\circ\text{C}$ or lower) see if one observes the S(VI)–F peak become a doublet. Also, H/F coupling would also show in the ^1H spectrum. This experiment would help to resolve:

- Distinguish between the $\text{RSO}_2[\text{DBU}]^+$ adduct and hydrogen bonding between SF and protonated DBU. If hydrogen bonding between SF and protonated DBU exists, one would observe H/F coupling both in the sulfonyl fluoride signal in the ^{19}F spectrum and in the ^1H NMR spectrum.
- If $\text{RSO}_2[\text{DBU}]^+$ adduct is present, you will see more consumption of the S–F signal in ^{19}F spectrum (using a fluorine based internal standard) and potentially the missing F^- peak.
- If more intermediate is formed and the experiments of S8/S9 were redone, it would be more convincing that the observed intermediate (whatever the speciation) is more likely involved in the reaction mechanism.
- In the absence of more evidence, it is challenging to the authors to eliminate any alternative mechanism or speciation.]

Our Response: We appreciate the reviewer’s suggestions, especially for the experimental design.

A number of NMR studies have been performed during this revision in response to the reviewer's concern. Here we will give a briefly summary of our conclusions based on the experimental results. The spectra and results are summarized in the revised SI (Page S72-S81) and are also appended in the next few pages (page 18-25).

1) hydrogen bonding of substrate ArSO₂F?

Table: A brief summary of the NMR experiments for hydrogen bonding study.

entry	NMR experiment	Spectra location (SI)	aim & conclusion
1	DBU : 1f = 2 : 1 (-40 °C -+20 °C, in CD ₃ CN)	Figure S13, S14 (SI, Page S73-S74)	Aim: to study the potential hydrogen bonding from -40 °C to 20 °C. Conclusion: 1) no hydrogen bonding with the ArSO ₂ F was found. 2) [FHF] ⁻ species was found below -20 °C, which was not observable at 20 °C (¹⁹ F NMR: δ -
2	DBU : TfOH : 1d = 1 : 1 : 1 (20 °C-+ -40 °C -+20 °C, in CD ₃ CN)	Figure S15, S16 (SI, Page S75-S76)	Aim: (DBU and TfOH are premixed to form a DBU•H ⁺ species quantitatively), and then to study if there it has any hydrogen bonding with the ArSO ₂ F. Conclusion: no [S-F-H] ⁺ bonding, but possible [S=O-H] ⁺ bonding with the ArSO ₂ F.
3	DBU : TfOH : 1d = 1 : 1 : 1 (20 °C-+ -60 °C -+20 °C, in CDCl ₃)	Figure S17, S18 (SI, Page S77-S78)	Aim: to rule out the solvent effect and temperature effect in studying the hydrogen bonding of ArSO ₂ F. (CD ₃ CN solvent might be a better hydrogen bond acceptor than ArSO ₂ F in previous NMR studies, making the above conclusion inaccurate. A weak coordinating solvent CDCl ₃ was used in this experiment. And the study was performed at a lower temperature) Conclusion: no [S-F-H] ⁺ bonding, but possible [S=O-H] ⁺ bonding with the ArSO ₂ F.
4	TfOH : 1d = 1 : 1 (20 °C-+ -60 °C -+20 °C, in CDCl ₃)	Figure S19, S20 (SI, Page S79-S80)	Aim: to detect potential hydrogen bond by simply mixing the acid with ArSO ₂ F in CDCl ₃ . Conclusion: no [S-F-H] ⁺ bonding, but possible [S=O-H] ⁺ bonding with the ArSO ₂ F.
1f was p -CF ₃ C ₆ H ₄ SO ₂ F, and 1d was PhSO ₂ F.			[1] J. Fluorine Chem. 2016 , 192 , 141-146

Our conclusion: We did not find any evidence of hydrogen bonding between the substrate ArSO₂F and the in situ formed acidic species when ArSO₂F and DBU were incubated in CD₃CN at a 1: 2 ratio. The existence of a [S=O-H]⁺ hydrogen bonding was possible between the intentionally prepared DBU•H⁺ species and the ArSO₂F (in CD₃CN and CDCl₃), and also between TfOH and ArSO₂F (in CDCl₃). In all these cases, a [S-F-H]⁺ hydrogen bonding was unlikely to be involved.

Considering that 1) the CH₃CN or DMSO solvents in the real reaction system were better hydrogen bonding acceptors than the ArSO₂F substrate, 2) the real reaction conditions was more basic because of excess DBU, and 3) the electron-deficient ArSO₂F substrates (poor hydrogen bonding acceptor) react much faster than the electron-rich ArSO₂F (good hydrogen bonding acceptor), there seems to be very little chance for a hydrogen bonding-induced reaction mechanism in our reaction.

(Note: Figure S13-S20 are appended on the next few pages)

Figure S13. ^1H NMR monitoring of the “**1f** (0.1 mmol) + DBU (0.2 mmol)” reaction mixture in CD_3CN (top: δ 16.4 – 1.0 ppm; bottom left δ 16.4 – 13.5 ppm, right δ 8.5 – 6.7 ppm)

Figure S14. ¹⁹F NMR monitoring of the “**1f** (0.1 mmol) + DBU (0.2 mmol) in CD₃CN” reaction mixture (top: δ 80.0 – -160.0 ppm; bottom left δ 65.5 – 67.7 ppm, middle 1 δ -62.5 – 64.5 ppm, middle 2 δ -108 – -110.5 ppm, right δ -143– -151.0 ppm.)

Figure S15. ¹H NMR monitoring of the potential hydrogen bonding between DBU·H⁺ and PhSO₂F in CD₃CN “DBU (0.1 mmol) + TfOH (0.1 mmol) + **1d** (0.1 mmol)” (δ 9.0 – 1.0 ppm)

65.5 65.4 65.3 65.2 65.1 65.0 64.9 64.8 64.7 64.6 -79.30
f1 (ppm)

S16-1: DBU (0.1 mmol) + TfOH(0.1 mmol)+ PhSO₂F in CD₃CN 20 °C
-79.40 -79.50 -79.60 -79.70
f1 (ppm)

Figure S16. ¹⁹F NMR monitoring of the potential hydrogen bonding between DBU•H⁺ and PhSO₂F in CD₃CN “DBU (0.1 mmol) + TfOH (0.1 mmol) + **1d** (0.1 mmol)” (top: 8 120 – -140 ppm; bottom: left 8 65.5 – 64.5 ppm, right 8 -79.3– -79.7 ppm.)

(Our analysis and conclusion: The ¹⁹F chemical shift changes of “-SO₂F” and “CF₃SO₂-” at different temperatures suggests possible [S=O–H]⁺ bonding with the ArSO₂F. The [S–F–H]⁺ bonding was ruled out because the split of “-SO₂F” signal by proton was not observed.)

Figure S17. ^1H NMR monitoring of the potential hydrogen bonding between $\text{DBU}\cdot\text{H}^+$ and PhSO_2F in CDCl_3 “ DBU (0.1 mmol) + TfOH (0.1 mmol) + **1d** (0.1 mmol)” (top: 8 10.0 – 1.0 ppm; bottom: left 8 9.0 – 7.2 ppm, right 8 6.7 – 4.2 ppm.)

(Our analysis and conclusion: The ^1H chemical shift changes of an active proton species (6.8 ppm – 4.2 ppm) at different temperatures suggests possible $[\text{S}=\text{O}-\text{H}]^+$ bonding with ArSO_2F . The $[\text{S}-\text{F}-\text{H}]^+$ bonding was ruled out because that the split of this proton peak by a fluorine was not observed.)

Figure S18. ¹⁹F NMR monitoring of the potential hydrogen bonding between DBU•H⁺ and PhSO₂F in CDCl₃ (“DBU (0.1 mmol) + TfOH (0.1 mmol) + **1d** (0.1 mmol)”) (top: δ 80.0 – 90.0 ppm; bottom: left δ 67.5 – 65.5 ppm, right δ -78.4– -78.9 ppm.)

(Our analysis and conclusion: The ¹⁹F chemical shift changes of “-SO₂F” and “CF₃SO₃H” at different temperatures suggests possible [S=O–H]⁺ bonding with the ArSO₂F. The [S–F–H]⁺ bonding was ruled out because that the split of the “-SO₂F” signal by a proton was not observed.)

Figure S19. ¹H NMR monitoring of the hydrogen bonding of TfOH and PhSO₂F in CDCl₃ (**1d** (0.1 mmol) + TfOH (0.1 mmol). (top: δ 12.5 – 6.5 ppm; bottom: left δ 12.2 – 11.3 ppm, right δ 10.9 – 9.3 ppm.)

(Our analysis and conclusion: The ¹H chemical shift changes of an active proton species (10.7 ppm – 9.5 ppm) at different temperatures suggests possible [S=O–H]⁺ bonding with the ArSO₂F. The [S–F–H]⁺ bonding was ruled out because that the split of this proton peak by a fluorine was not observed.)

Figure S20. ¹⁹F NMR monitoring of the hydrogen bonding of TfOH and PhSO₂F in CDCl₃ (**1d** (0.1 mmol) + TfOH (0.1 mmol). (top: 8 90.0 – -90.0 ppm; bottom: left 8 67.0 – 65.0 ppm, right 8 -75.7 – -77.0 ppm.)

(Our analysis and conclusion: the ¹⁹F chemical shift changes of “-SO₂F” and “CF₃SO₃H” at different temperatures suggests possible [S=O–H]⁺ bonding with the ArSO₂F. The [S–F–H]⁺ bonding was ruled out because that the split of the “-SO₂F” signal by a proton was not observed.)

2) accumulation of the **Int** species?

We have separately performed the reactions between **1f** and DBU in CD₃CN at different temperatures. The NMR spectra are summarized below. And the data analysis chart is appended on the next page.

Figure S21. ¹H NMR monitoring of the accumulation of the **Int** species “**1f** (0.1 mmol) + DBU (0.2 mmol) in CD₃CN” (top: δ 8.5 – 1.3 ppm; bottom: δ 8.4 – 6.6 ppm.)

Figure S22. Study the accumulation of **Int** species at different temperatures.

Our analysis and conclusion: 1) the yield of the **Int** species did not increase with times at room temperature (S21-1, S21-2). A by-product was formed after 5 hours of reaction at room temperature. We could not identify its chemical structure (S21-2). 2) yields of the **Int** species were relatively lower at low reaction temperatures (S21-3 to S21-5). And we did not find the by-product at low reaction temperatures even with longer reaction time, which was observed at room temperature.

Response to Reviewer 3:

Reviewer 3: Ligation chemistry of SFs (sulfonyl fluorides and sulfonimidoyl fluorides) as a result of fluorine substitution is unique. And sulfur(VI) fluoride exchange (SuFEx) based on SFs has recently emerged as a valuable tool for different synthetic purposes (Ref 15, Sharpless, *Angew. Chem. Int. Ed.*, 2014, 53, 9430). The manuscript by Wu and Gao details extensive work in the field of SuFEx ligations through a different approach. It is true that nucleophilic substitution pathways have been exclusively practiced in previous use of SFs while radical process is challenging and little explored. Widened utility of SFs through radical ligation with the method described in this paper would be a big step forward.

The paper outlines the reaction design and development. With cooperative activation by

organo-superbase and photoredox catalysis, electrophilic SFs are harnessed for radical functionalization of styrene derivatives. The mechanism looks good in supporting the proposed reaction pathway. It is interesting that DBU plays an essential role. This dual activation model should be inspiring for future development of strategies in activating other inert bonds.

The authors demonstrate synthetic applications of their method and products derived therefrom. Given the wide availability of SFs and mild ligation condition, this method should be useful to the synthetic community and beyond. The unsymmetric divinyl sulfones are attractive Michael acceptors either in bioconjugation or synthetic materials.

In summary, the paper provides a nice extension of the SuFEx ligation. The method is useful and the strategy meaningful. SFs are not simple analogues of sulfonyl chlorides or sulfoximidoyl chlorides (SCls, which are known for radical reactions). They are doing much better than SCls in terms of stability-reactivity balance. The development of radical ligation based on SFs is more challenging but will be paid off. The work satisfies the novelty requirements necessary for Nature Communication. I strongly recommend publication of this paper.

Additional comments:

1) Substrate scope of the sulfur(VI) radical acceptor is relatively narrow and limited to styrene derivatives. Similar S-radicals like Ts radical could react with a wide range of alkenes. Could the authors comment about this? Would it be possible to switch to other alkenes such as acrylate derivatives and even unactivated alkenes?

Our Response: We thank the reviewer for the valuable feedback. Yes, one limitation of this method so far is that only the styrene derivatives work. Acrylates and unactivated alkenes have failed (ArSO₂F partially decomposed instead of radical coupling with alkenes in these reactions). The reason is not clear yet. We hypothesized that it is probably because of the mismatched redox potentials of the related intermediates. The corresponding discussions can be found in the revised manuscript (Page 3, right column, lines 28-30).

Reviewer 3: 2) Post-transformations in Figure 3C are useful but might not be well-presented in terms of clarity. This part should be improved.

Our Response: Thanks. We have combined Figure 3B and 3C and reorganized the related contents.

B. grams scale synthesis and Single molecular post-modification

[a] trimethylsulfoxonium iodide, NaOH, rt; [b] tris(trimethylsilyl)silane, AIBN, reflux; [c] ethyl isocynoacetate, NaOH, rt; [d] DPPPO, KOH, O₂, rt; [e] benzyl mercaptan, Et₃N, rt; pyrrolidine, r1; [g] dimethyl malonate, triton-B, 70 °C; [h] cyclopentanone, triton-B, 70°C; [i] n-butylamine, 70 °C

Reviewer 3: 3) There is another paper online very recently that deal with radical sulfonylation with aryl sulfonyl fluorides (doi.org/10.1021/acs.orglett.3c00447). I guess these two independent papers were completed and submitted almost at the same time. Synthetic method described in this paper looks strategically different and seems more general, as it also applies to vinyl sulfonyl fluorides and sulfonimidoyl fluorides (these two types of substrates are not discussed in the OL paper, probably not working). Therefore, the novelty and significance of this paper would not be affected. But the OL paper should be cited in the revised manuscript.

Our Response: Thanks. The Luo and Molander groups provide elegant examples of activating aryl sulfonyl fluorides for the sulfonylation reactions through radical intermediates. Luo's work was cited as Ref 51 in the previous manuscript. Molander's work was published online after the submission of this manuscript. In the revised manuscript, both papers have been cited (Refs 49,50). The related discussions have also been added at the end of paragraph 3 of the Results and Discussion section.

Reviewer 3: 4) Does this method work for sulfonyl chlorides? It would be interesting to compare the results of these two sulfonyl sources.

Our Response: We have tried $p\text{-CH}_3\text{C}_6\text{H}_4\text{SO}_2\text{Cl}$ as the substrate (Table 1, entry 16, in the manuscript). It decomposed under our standard reaction conditions, giving only trace amount of the desired product. According to our experience, sulfonyl chlorides are susceptible to hydrolysis in the presence of DBU. By the way, we have also studied the reaction of sulfonimidoyl chloride. Its reduction to sulfenamide is the major reaction pathway.

Reviewer 3: 5) In the proposed mechanism, a carbocation intermediate is formed. Thus there is an opportunity to develop alkene difunctionalization reactions through trapping the cation intermediate with nucleophiles.

Our Response: We thank the reviewer for the nice suggestion. We agree and have actually investigated a few reactions so as to interrupt the reaction by trapping the benzylic cation intermediate formed in situ. However, we have not found any positive results yet.

REVIEWERS' COMMENTS

Reviewer #1 (Remarks to the Author):

According to the peer review, the authors have addressed all the concerns from my previous review. The positive revisions have been provided in this version. Notably, the relevant reasons of low efficiency for the S–C(sp²) ligation of N-Ts, N-alkyl, and N-aryl sulfonimidoyl fluorides have been elucidated in the revised manuscript. The key intermediate has been confirmed adequately via control experiments and NMR spectra. In a word, these revisions meet the requirements and I recommend this manuscript for publication.

Two suggestions:

- a) The reference (Natl. Sci. Rev. 2023, 10.1093/nsr/nwad123.) about the SuFEx linkers should be added in Refs 19-21.
- b) The content of NMR study should be clear. Some items can be placed in the note of figure, and the text should not overlap with the spectrum.

Reviewer #2 (Remarks to the Author):

The revised manuscript – NCOMMS-23-11185A – by Gao and et. demonstrated significant improvement. The authors should be commended for their additional work and detail especially regarding the reviewer comments/suggestion on the proposed mechanism. The authors successfully addressed the reviewers concerns and thus my recommendation is for publication.

Reviewer #3 (Remarks to the Author):

The quality of the manuscript has been improved compared to the previous version. The reviewer's comments and questions were addressed in the revision and I am satisfied with the responses.

Response to Reviewer 1:

According to the peer review, the authors have addressed all the concerns from my previous review. The positive revisions have been provided in this version. Notably, the relevant reasons of low efficiency for the S–C(sp²) ligation of N-Ts, N-alkyl, and N-aryl sulfonimidoyl fluorides have been elucidated in the revised manuscript. The key intermediate has been confirmed adequately via control experiments and NMR spectra. In a word, these revisions meet the requirements and I recommend this manuscript for publication.

Two suggestions:

a) The reference (Nat. Sci. Rev. 2023, 10.1093/nsr/nwad123.) about the SuFEx linkers should be added in Refs 19-21.

Our Response: We thank the reviewer for kind suggestion. The reference (*Nat. Sci. Rev.* **2023**, 10.1093/nsr/nwad123.) has been cited as Ref 22.

b) The content of NMR study should be clear. Some items can be placed in the note of figure, and the text should not overlap with the spectrum.

Our Response: The NMR spectra of Figure 6C (previous Figure 2C) has been modified. And some NMR spectra in Supplementary Information have also been changed (page 67-81).

Response to Reviewer 2:

The revised manuscript – NCOMMS-23-11185A – by Gao and et. demonstrated significant improvement. The authors should be commended for their additional work and detail especially regarding the reviewer comments/suggestion on the proposed mechanism. The authors successfully addressed the reviewers concerns and thus my recommendation is for publication.

Our Response: We thank the reviewer for helpful discussion.

Response to Reviewer 3:

The quality of the manuscript has been improved compared to the previous version. The reviewer's comments and questions were addressed in the revision and I am satisfied with the responses. **Our**

Response: We thank the reviewer for helpful discussion.